# Free circular introns with an unusual branchpoint in neuronal projections

Harleen Saini[1,2]*, Alicia A Bicknell[1†], Sean R Eddy[2,3]*, Melissa J Moore[1†]*

[1]RNA Therapeutics Institute, University of Massachusetts Medical School, Worcester, United States; [2]Department of Molecular and Cellular Biology, Howard Hughes Medical Institute, Harvard University, Cambridge, United States; [3]John A Paulson School of Engineering and Applied Sciences, Harvard University, Cambridge, United States

**Abstract** The polarized structure of axons and dendrites in neuronal cells depends in part on RNA localization. Previous studies have looked at which polyadenylated RNAs are enriched in neuronal projections or at synapses, but less is known about the distribution of non-adenylated RNAs. By physically dissecting projections from cell bodies of primary rat hippocampal neurons and sequencing total RNA, we found an unexpected set of free circular introns with a non-canonical branchpoint enriched in neuronal projections. These introns appear to be tailless lariats that escape debranching. They lack ribosome occupancy, sequence conservation, and known localization signals, and their function, if any, is not known. Nonetheless, their enrichment in projections has important implications for our understanding of the mechanisms by which RNAs reach distal compartments of asymmetric cells.

*For correspondence:
harleen.saini@umassmed.edu
(HS);
seaneddy@fas.harvard.edu (SRE);
melissa.moore@umassmed.edu
(MJM)

Present address: †Moderna
Therapeutics, Cambridge, United
States

Competing interests: The
authors declare that no
competing interests exist.

Reviewing editor: Benjamin J
Blencowe, University of Toronto,
Canada

## Introduction

In polarized cells, such as neurons and oocytes, RNA localization to distinct subcellular compartments is important for spatial control of protein expression (*Holt and Bullock, 2009*). Known mechanisms for asymmetric distribution of RNA include active transport (e.g., *Actb*; *Ross et al., 1997*), spatially restricted capture by an anchor (e.g., *Nanos*; *Forrest and Gavis, 2003*), and control of RNA degradation (e.g., *Hsp83*; *Bashirullah et al., 2001*). In some cases, RNA localization depends on splicing. For example, a detained intron (i.e., an intron with regulated post-transcriptional splicing, as opposed to constitutive co-transcriptional splicing) restricts *Srsf5* mRNA export from the nucleus (*Boutz et al., 2015*), and a retained intron (i.e., an alternative unspliced isoform) promotes dendritic localization of *Calm3* (*Sharangdhar et al., 2017*). The *Robo3* gene, which is important for commissural axon development in mice, expresses both a fully spliced mRNA and another retaining intron 26, and these isoforms encode different proteins that have opposing functions in axon guidance (*Chen et al., 2008*). Spatial and temporal control of protein expression from the intron-retaining *Robo3* isoform depends on its susceptibility to nonsense-mediated decay due to the presence of a premature termination codon in the retained intron (*Colak et al., 2013*).

Provocatively, some retained introns have been proposed to undergo splicing in dendrites (*Glanzer et al., 2005*). For example, an intron in the calcium-activated potassium channel *Kcnma1* was reported to undergo splicing in dendrites of rat hippocampal neurons (*Bell et al., 2010*), and this was suggested to be a mechanism for locally tailoring calcium-activated potassium currents. Because pre-mRNA splicing by the spliceosome is generally thought to be restricted to the nucleus (*Steitz et al., 2008*), this proposal has been controversial, and it has not yet been independently confirmed.

The interplay between intron retention and neuronal RNA localization has been studied in several individual cases (*Chen et al., 2008*; *Bell et al., 2010*; *Buckley et al., 2011*; *Khaladkar et al., 2013*;

*Ortiz et al., 2017*; *Sharangdhar et al., 2017*). In this work, our aim was to systematically identify localized RNAs in primary rat hippocampal neurons by sequencing total RNA (rRNA depleted) as opposed to polyadenylated (polyA+) RNA, with a particular focus on the repertoire of projection-localized introns (both retained and excised). Our analyses identify hundreds of transcripts with retained introns. Unexpectedly, we also found a set of free circular introns localized to distal neuronal projections.

## Results

### Experimental design and validation

To physically separate cellular projections from cell bodies, we cultured dissociated primary rat hippocampal cells on membranes with 1 μm diameter pores (*Poon et al., 2006*). These cultures are a mixture of neuronal and glial cells; we add a DNA replication inhibitor to block cell division and prevent dividing glia from overgrowing post-mitotic neurons. We refer to the projections as 'neuro-glial' projections because both neuronal (Map2-immunopositive) and non-neuronal (Gfap/Vimentin-immunopositive) projections extend through the pores and continue growing on the underside of the membrane, whereas cell bodies and nuclei are restricted to the top surface (*Figure 1A* and *Figure 1—figure supplement 1*). Lysates prepared by scraping the underside are highly enriched for projections ('projection' samples), while lysates prepared from the top surface comprise whole cells with nuclei and projections ('whole cell' samples).

To capture both adenylated (polyA+) and non-adenylated (polyA-) long RNAs in our lysates, we prepared rRNA-depleted total RNAseq libraries (mean insert size ~200 nt) from five biological replicates (ten samples total). The RNAseq libraries were subjected to paired-end sequencing (100–125 nt reads) on the Illumina platform to obtain 30–80 million mate pairs per sample.

We generated additional datasets to help interpret projection and whole cell RNAseq data. To distinguish retained introns from intronic polyadenylation (polyA) sites, we prepared polyA-site sequencing (PASseq) libraries (*Ashar-Patel et al., 2017*) from a subset of samples (three biological replicates each, six samples total). To identify coding exons, we generated ribosome profiling (*Ricci et al., 2014*) and polyA+ selected RNAseq libraries (*Heyer et al., 2015*) from the purified cytoplasmic (nuclei-depleted) fraction of primary rat hippocampal neurons cultured on plates (three biological replicates) (*Figure 1A and B*; *Figure 1—source data 1*).

For preliminary characterization of these data, we aligned RNAseq reads to the rat genome (Ensembl release 81, Rnor_6.0) (*Zerbino et al., 2018*) (*Figure 1—figure supplement 3*). Ribosomal RNAs and small (<150 nt) noncoding RNAs (e.g., snRNAs, tRNAs) were depleted effectively. Larger abundant nonpolyadenylated noncoding RNAs were observed as expected, such as SRP (or 7SL, the RNA component of the signal recognition particle) and RMRP (the RNA component of mitochondrial RNase P); both were more prevalent in projection than in whole cell libraries. Reads mapping to the mitochondrial genome were also highly abundant and enriched in projection libraries, consistent with the abundance of mitochondria in neurons and neuronal projections (*Palay, 1956*) (*Figure 1—figure supplement 1*). To minimize spurious genome alignments due to expected abundant species, we filtered out reads mapping entirely to rRNA, SRP, mitochondrial genome, and repeat elements cataloged by RepeatMasker (*Jurka, 2000*) before moving on to other analyses described below.

To qualitatively assess our success in separating projections from whole cells, we aligned filtered reads to the rat genome with TopHat2 (*Kim et al., 2013*) and visualized read density tracks of known nuclear and projection localized RNAs on a rat genome browser (*Robinson et al., 2011*). Known nuclear-localized RNAs such as the noncoding RNA *Xist* (*Brown et al., 1992*) and *Srsf5* mRNA (which contains a detained intron) (*Boutz et al., 2015*) were depleted from projections relative to whole cells (*Figure 1C* and *Figure 1—figure supplement 1*). Conversely, the known projection-localized *Pabpc1* mRNA (*Poon et al., 2006*) was enriched in projection data.

### RNAs enriched in projections

To comprehensively and quantitatively evaluate how well our datasets distinguish known localized RNAs, we employed Kallisto and Sleuth (*Bray et al., 2016*; *Pimentel et al., 2017*) for differential expression analysis of annotated RNA transcripts in projections versus whole cells. RNA abundances (in TPM, Transcripts Per Million) of biological replicates were well correlated (Spearman's correlation

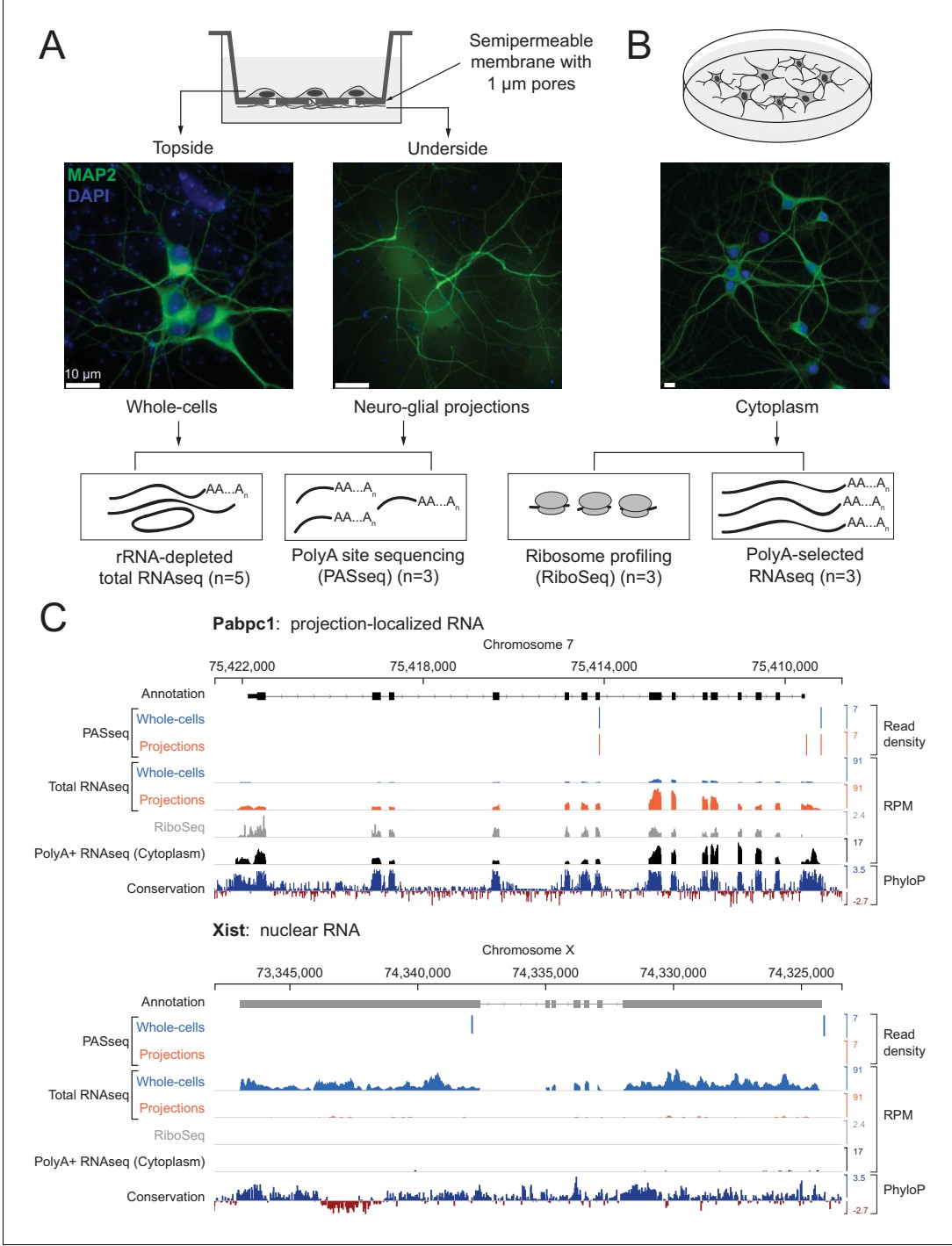

**Figure 1.** Experimental design and data validation. (**A**) Imaging of MAP2 protein immunostaining (neuronal marker, green) and DAPI fluorescence (nuclear marker, blue) confirms that the bottom surface of neuronal cultures on a semipermeable membrane (cartoon, top) consists only of neuro-glial projections. Total RNAseq and polyA site RNAseq (PASseq) datasets were generated from the top surface ('whole cell') and bottom surface ('projection') lysates. (**B**) Standard plate cultures, fractionated to remove nuclei, were used to prepare ribosome profiling and cytoplasmic polyA+ RNAseq datasets. (**C**) Genome browser plots of read densities (sum of three replicates) and sequence conservation (PhyloP scores on 20 aligned vertebrate genomes, *Pollard et al., 2010*) for a projection-localized mRNA and a nuclear noncoding RNA (RPM = Reads per million mapped).

The online version of this article includes the following source data and figure supplement(s) for figure 1:

**Source data 1.** Number of reads sequenced and alignment statistics for each library.

*Figure 1 continued on next page*

*Figure 1 continued*

**Figure supplement 1.** Separation of neuro-glial projections from cell bodies.
**Figure supplement 2.** Microcapillary electrophoresis of total RNA from whole cells and projections.
**Figure supplement 3.** Data analysis workflow.

coefficient ≥ 0.83 in projection samples, ≥ 0.88 in whole cell) (*Figure 2—figure supplement 1*), but comparisons between projection and whole cell datasets showed substantial differences (*Figure 2A*; *Figure 2—source data 1*). As expected, known nuclear noncoding RNAs, including *Xist*, *Malat1*, *Meg3*, snoRNAs, and scaRNAs, were among the 1486 genes significantly depleted (>1.5 fold and q-value <0.01) from projections. In contrast, 1440 transcripts were significantly enriched in projections, including known projection-localized mRNAs such as *Pabpc1*, *Map2*, *Dlg4* (neuronal), and

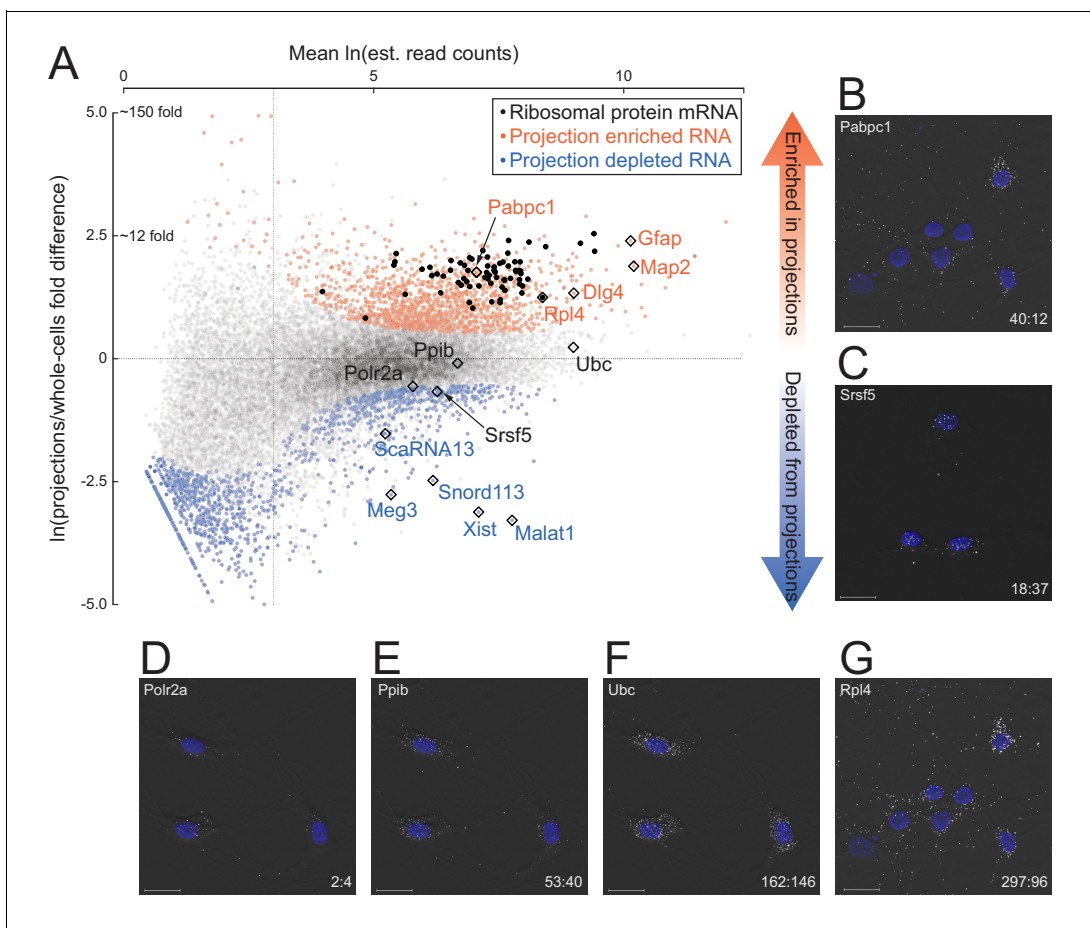

**Figure 2.** Quantitative analysis of RNA localization. (**A**) Scatter plot comparing log mean read counts and log fold difference in projections versus whole cells, for 19,815 RNA transcripts with non-zero read counts. 1440 (orange dots) are significantly enriched (*q*-value < 0.01 and fold-change > 1.5) in projections, 1486 (blue dots) are enriched in whole cells, and 16,899 (gray dots) show no significant enrichment in either sample. Ribosomal protein encoding RNAs are shown as black dots; rhombi enclose labeled RNAs. (**B–G**) smFISH validation of examples of projection-localized (**B**) versus projection-depleted (**C**), and low (**D**), medium (**E**), high (**F**), and higher (**G**) abundance mRNAs. Upper left corner shows gene name, lower right corner shows projection:whole cell TPM. Scale bars = 20 µm. The online version of this article includes the following source data and figure supplement(s) for figure 2:

**Source data 1.** Isoform abundances and differential expression analysis output.
**Source data 2.** Sequences targeted by smFISH probes.
**Figure supplement 1.** Correlation between RNAseq from biological replicates.
**Figure supplement 2.** Gene ontology classes enriched in projections and whole cells.

*Gfap* (glial) (*Poon et al., 2006*; *Garner et al., 1988*; *Cajigas et al., 2012*; *Sarthy et al., 1989*). Gene ontology analysis showed that the set of projection-enriched mRNAs were significantly enriched for genes involved in mitochondrial functions (cytochrome-c oxidase activity) as well as nearly the entire set of ribosomal protein encoding mRNAs (RP mRNAs) (*Figure 2—figure supplement 2*). Indeed, seventy annotated RP mRNA isoforms were enriched more than two-fold in projections (black dots in *Figure 2A*). RP mRNA enrichment in projections is puzzling because ribosomal proteins are imported into the nucleus for ribosome assembly but their enrichment in distal cellular locations has been made consistently in other RNA localization studies (for e.g., most recently, *Shigeoka et al., 2018* studied RP mRNA enrichment in *Xenopus* retinal cell axons), and we discuss it later.

To test the accuracy of our RNAseq abundance measurements, we performed single-molecule fluorescence in situ hybridization (smFISH) on example RNAs. Using exon-hybridizing RNAScope probe sets, we probed for six different mRNAs spanning a range of TPM values and projection: whole cell ratios (*Polr2a*, 2:4; *Ppib*, 43:50; *Ubc*, 162:146; *Pabpc1*, 40:12; *Srsf5*, 18:37; *Rpl4*, 297:96) (*Figure 2B–G*; *Figure 2—source data 2*). Consistent with their RNAseq projection:whole cell TPM ratios, *Polr2a*, *Ppib*, and *Ubc* mRNAs exhibited predominantly cytoplasmic localization, with a strong gradient in spot density highest in cell bodies tapering off into the projections. *Srsf5* mRNA was almost entirely confined to the cell bodies, and consistent with the presence of a detained intron (*Boutz et al., 2015*), *Srsf5* mRNA spots were also visible in the nucleus. In contrast, *Pabpc1* and *Rpl4* mRNA spots exhibited a nearly uniform distribution throughout cell bodies and projections, with relatively few discernible spots in nuclei.

We concluded from these and other analyses that our RNAseq datasets reliably detect and quantify projection-localized RNAs in primary rat hippocampal cultures. We then turned to our main interest in localization of intron sequences.

## Intron regions enriched in projections

Because of alternative splicing, transcribed genomic regions cannot be easily separated into introns and exons. To facilitate a comprehensive analysis, we define an 'intron region' as a genomic interval that is annotated as intronic (and not exonic) in all annotated transcript isoforms that span it (Ensembl release 81, Rnor_6.0, annotation downloaded on July 24, 2015) (*Figure 3A* and *Figure 3—figure supplement 1*). Out of a total of 190,180 such intron regions, we considered 57,432 to have reliable coverage (at least one read in each of the five biological replicates, with mean read density > 0.005 mapped reads/intron region length) in the whole cell libraries, but only 1632 met these criteria in projections (*Figure 3B* - inset; *Figure 3—source data 1*). For the 33 intron regions that we considered reliably covered in projections but not in whole cells, individual examination showed that all had coverage in the whole cell libraries but had just missed the cut.

Introns are expected to be spliced and degraded in the nucleus and thus strongly depleted from projections, but a scatter plot of intron region abundance in projections versus whole cells (*Figure 3B*) shows a bimodal distribution. Intron regions that show coverage in projections define a subpopulation that has similar read coverage in projections and whole cells. This population is interesting because of the restrictive way that we define 'intron regions': no annotated transcript isoform of a given gene shows these regions as exonic, but their abundance in projections suggests that they are either unspliced, excised but stable, or independent transcripts. They include, for example, known (but unannotated) neuron-specific retained introns, such as in *Calm2* (homolog of mouse *Calm3*; *Sharangdhar et al., 2017*).

As we looked at randomly selected examples of the 1632 intron regions in the rat genome browser, we found, unsurprisingly, that many cases simply represented an unannotated alternative splicing event (i.e., alternative 5' or 3' splice site) or an unannotated transcription start site (TSS) or polyadenylation site (PAS) within an annotated intron. We found it useful to distinguish them into classes depending on the presence or absence of reads spanning the unspliced 5' or 3' splice sites (5' exon-intron and intron-3' exon reads, EI and IE) and spliced exon-exon junctions (EE) (*Figure 4A, B* and *Figure 4—figure supplement 1*).

Intron regions with high EI reads but few IE reads (*Figure 4B–(i) upper left; n* = 385 regions) correspond to unannotated alternative 5' splice sites (e.g., *Aplp1*) or unannotated alternative polyadenylation sites (e.g., *Cxadr*). Conversely, regions with high IE but few EI reads (*Figure 4B-(iii) lower right; n* = 320 regions) correspond to unannotated alternative 3' splice sites (e.g., *Mtss1l*) or unannotated alternative transcriptional start sites (e.g., *Nell2*).

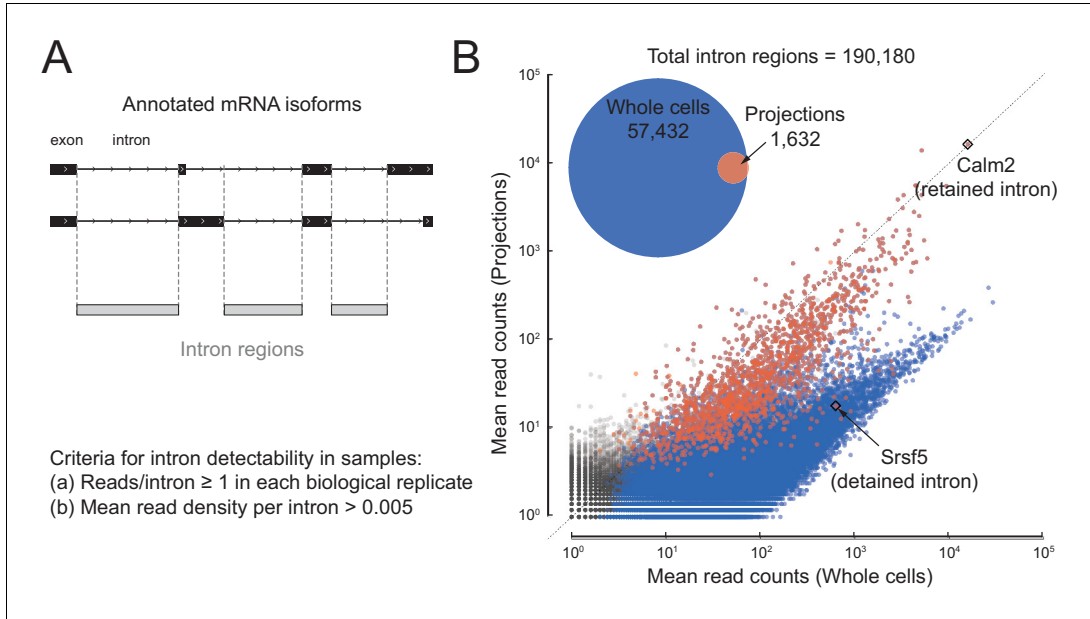

**Figure 3.** A subset of introns localize to projections. (**A**) Cartoon illustrating how we define intron regions as the intersection of all annotated introns. For two mock alternatively spliced isoforms, black rectangles are exons, lines are introns; vertical dotted lines mark the intron region boundaries. (**B**) Inset: Venn diagram showing the number of intron regions that we consider to show reliable read coverage (at least one read in each replicate and mean read density > 0.005 reads/nucleotide). Scatter plot comparing mean mapped reads per intron region (+1 pseudocount) from five biological replicates of projections versus whole cell RNAseq data. Out of 190,180 intron regions (gray), 57,432 pass the detectability threshold in whole cell (blue overlay) and 1632 in projection (red overlay) libraries. Rhombi enclose labeled RNAs.
The online version of this article includes the following source data and figure supplement(s) for figure 3:

**Source data 1.** Compilation of observations on all 190,180 intron regions.
**Figure supplement 1.** Data analysis workflow to count reads on intron regions.
**Figure supplement 2.** Comparison with previously reported retained introns in dendrites.

---

High EI and IE reads correspond predominantly to retained introns (*Figure 4B-(ii) upper right*; n = 428 regions), the most abundant of which are a 3' UTR intron in *Calm2* and the last intron in *Sept3* (intron 10). Previous work in mouse neurons showed that the dsRNA binding protein STAU2 interacts with the *Calm3* intron to promote dendritic localization of the intron-retaining mRNA isoform (*Sharangdhar et al., 2017*). In our data, the intron-retaining isoform of *Calm2* predominated in both projection and whole cell libraries, with no selective enrichment in either cell compartment (*Figure 4—figure supplement 2*). Retention of *Sept3* intron 10 results in a protein isoform with a different C-terminal sequence than the canonical isoform. Conservation of this alternative coding sequence suggests it is likely functional (*Figure 4C*).

We looked specifically at *Kcnma1* intron 23, which has previously been reported to be retained, localized to rat primary hippocampal neuron projections (*Bell et al., 2008*), and spliced locally in dendrites upon neuronal activation (*Bell et al., 2010*). In our data, no *Kcnma1* introns appear to be retained or localized to projections (*Figure 4—figure supplement 3*). We detect only spliced *Kcnma1* transcript isoforms in projections.

A fourth class of intron regions in projections had both low EI and IE reads (*Figure 4B-(iv) lower left*; n = 499 regions). Some of these regions proved to harbor a gene transcribed from the same strand and contained within the intron of a different gene, such as *Cox6a1* within the last intron of *Gatc* (*Figure 4—figure supplement 1*). Others corresponded to unannotated alternative terminal exons, as in *Map4* (*Figure 4—figure supplement 1*). We used the presence of intronic polyadenylation sites from our PASseq data to identify these two cases (n = 96 combined). Another subset corresponded to unannotated alternative cassette exons, as in *Abi2* (*Figure 4—figure supplement 1*), which we identified using evidence of ribosome occupancy (mean ribosome profiling reads ≥5 per

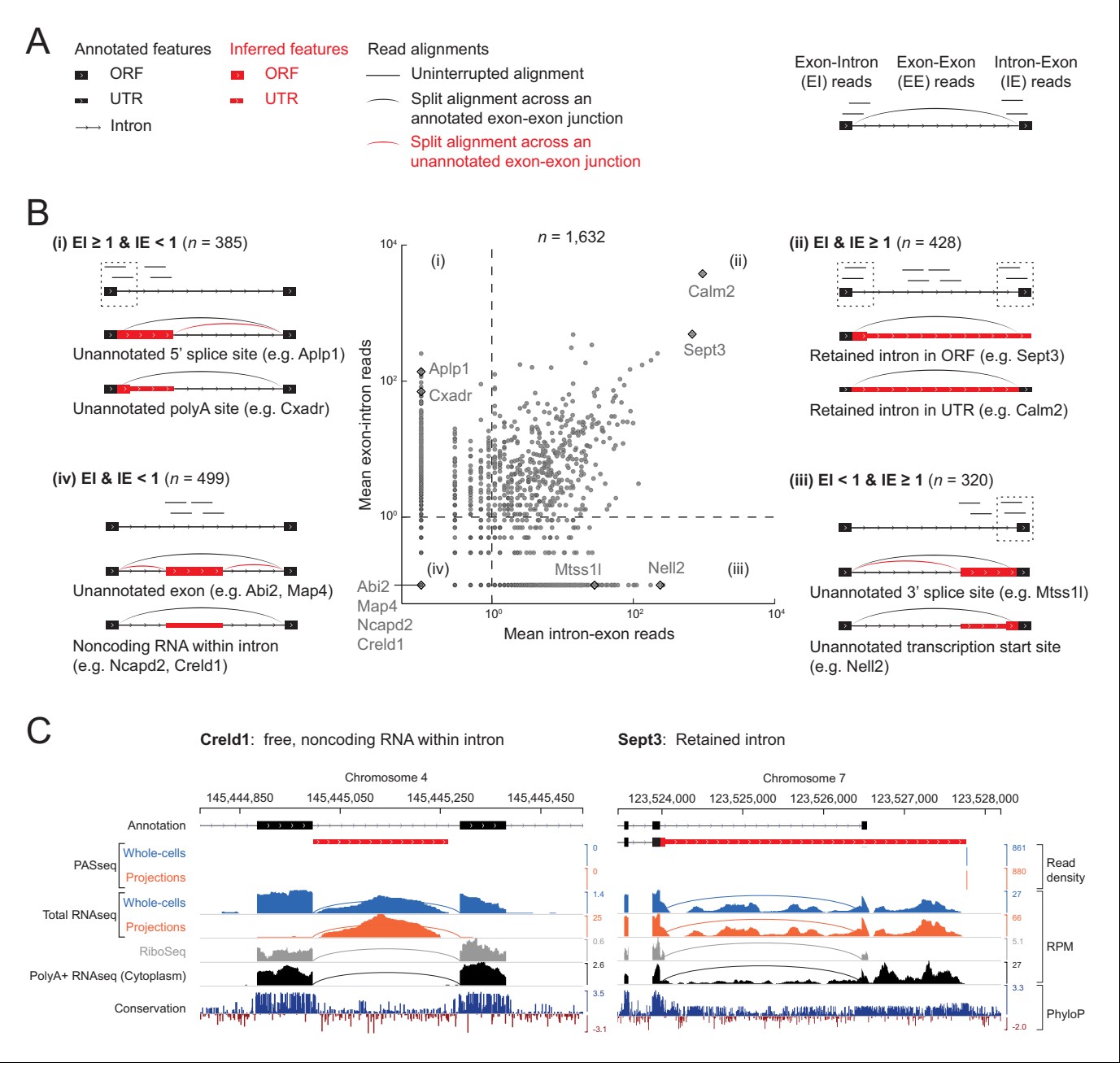

**Figure 4.** Classification of 1632 projection-localized intron regions. (**A**) Description of symbols used to show annotated gene structure and types of read alignments. Rectangles represent annotated (black) or inferred (or unannotated, red) coding (ORF, Open Reading Frame) and noncoding (UTR, UnTranslated Region) exons. Lines represent introns and arrows point in the 5' to 3' direction of the transcript. Uninterrupted read alignments are shown as lines whereas arcs depict split reads connecting spliced exon-exon junctions. (**B**) Scatter plot of mean read coverage (+0.1 pseudocount) of 50 nt exon-intron (EI) versus intron-exon (IE) boundary regions in projection RNAseq samples. Dashed lines indicate thresholds set to EI = 1 and IE = 1 that divide introns into four quadrants (**i-iv**). Representative sketches depicting the situations within each quadrant are shown. *n* shows the number of intron regions in each quadrant. (**C**) Genome browser views of read coverage (sum of three biological replicates) and phyloP conservation for a free (Creld1-intron 5) and retained intron (Sept3 - 3' terminal intron). RPM = reads per million mapped.

The online version of this article includes the following figure supplement(s) for figure 4:

**Figure supplement 1.** Genome browser views of introns labeled in *Figure 4*.

**Figure supplement 2.** Comparison of intron enrichment versus mRNA enrichment in projections.

**Figure supplement 3.** Genome browser view of *Kcnma1* shows only the spliced isoform in projections.

**Figure supplement 4.** Subcellular location of constitutively spliced, retained, detained, and free introns validated by smFISH.

replicate; *n* = 221). The remaining 278 regions showed no evidence of ribosome occupancy in our ribosome profiling data, nor of polyadenylation sites in our PASseq data, and thus they appeared to correspond to nonpolyadenylated noncoding RNAs – possibly free introns or other genes within an intron – and we looked into them further.

## Circular introns with noncanonical branchpoints in projections

We validated the localization of a detained intron (Srsf5-intron5), retained intron (Sept3-intron10), and free intron (Creld1-intron4) by smFISH (*Figure 4—figure supplement 4*). As a negative control, we examined the localization of constitutively spliced introns in *Srsf5* and *Sept3* by smFISH. As expected, the constitutively spliced introns in *Srsf5* and *Sept3* were observed only in the nucleus, as was the detained intron albeit at a higher copy number. The retained intron of *Sept3* could be detected in the projections. Dual color smFISH with probes targeting the exons or retained intron of *Sept3* shows close proximity of the signal from the exons and introns. In the case of the free intron from *Creld1*, the intron signal could be detected in both nuclei and distal projections. Outside the nucleus, no colocalization was observed between free intron and exon signal from *Creld1*.

One explanation for detecting 'free' intron regions in projection samples is that they could correspond to intron-encoded small RNAs such as small nucleolar RNAs (snoRNAs) or small Cajal body RNAs (scaRNAs) that are nuclear-localized yet abundant enough that we detect reads in both projection and whole cell data. We plotted relative enrichment (log ratio in projections/whole cells) versus

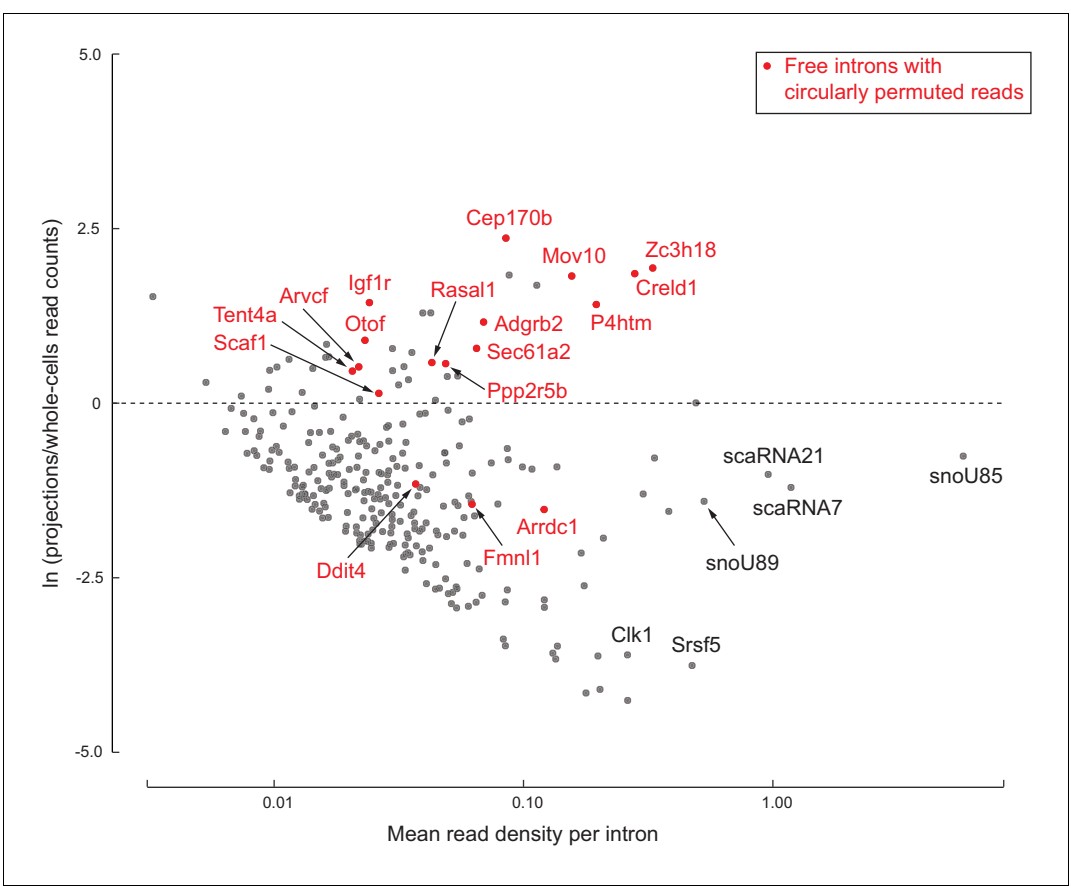

**Figure 5.** 278 free introns detected in projections. Scatter plot of log ratio of mean read density per intron region in projections versus whole cells, and read density averaged across all replicates of projections and whole cells. Examples of previously known detained introns and intronic snoRNAs/scaRNAs are labeled in black. Introns with circularly permuted reads and without known functional annotation are labeled in red.

The online version of this article includes the following figure supplement(s) for figure 5:

**Figure supplement 1.** Genome browser plots of projection-localized free introns with circularly permuted reads.

average abundance of each of the 278 intron regions (*Figure 5*). This showed that most regions are indeed depleted in projection data, and we found that many do correspond to known snoRNAs and scaRNAs. Projection-depleted regions also included known detained (unspliced and nuclear-localized) introns of highly expressed genes, including *Srsf5* and *Clk1*. We attribute their low EI and IE read counts in projection data to statistical fluctuation around our thresholds; in whole cell data, we see high EI and IE read counts for these abundant nuclear-localized introns.

However, many other intron regions were both abundant and enriched in projections (*Figure 5*). These have features indicative of a previously described class of RNAs called 'stable intronic sequence RNAs' (sisRNAs; *Gardner et al., 2012*). SisRNAs are circular lariat products of splicing (i. e., free introns) that are inefficiently debranched in the nucleus and exported to the cytoplasm via an NXF1/NXT1-dependent mechanism (*Talhouarne and Gall, 2018*). The most prominent examples in our projection libraries, such as *Creld1* (*Figure 4C* and *Figure 5*), *Zc3h18*, and *Mov10*, were devoid of exon-intron and intron-exon reads, with higher read density on the intron than on the flanking exons. Further, these species were not detected in polyA+ RNAseq and ribosome profiling data. We also observed a lack of read coverage over a 20–30 nt region at the 3' end of the intron (*Figure 5— figure supplement 1*).

Spliceosome-mediated intron excision from pre-mRNA releases a lariat molecule in which the branchpoint nucleotide, predominantly an adenosine (*Taggart et al., 2017*), is linked 2'−5' to the 5' end of the intron. Reverse transcriptase can occasionally traverse a 2'−5' linkage, so lariat branchpoints result in circularly permuted reads in RNAseq data. To test whether projection-enriched free intron species had characteristics of an intron lariat, we searched for circularly permuted reads within them using find_circ.py (*Memczak et al., 2013*). For fourteen projection-enriched intron regions, we detected numerous circularly permuted reads (*Figure 6A*), enabling us to identify the branchpoints. In all 14 cases, the branchpoint was a cytosine, C ($n$ = 12), or guanine, G ($n$ = 2), instead of the canonical adenosine, A (*Figure 6—source data 1*). The spliceosome can use C or G as a branchpoint nucleotide, but the lariat debranching enzyme is inefficient at hydrolyzing the 2'−5' linkage at these residues (*Jacquier and Rosbash, 1986*). Thus lariats with C or G branchpoints might be expected to be more stable than other introns. The lack of read coverage at the 3' end of these introns could arise from failure to clone the short lariat tails (< 50 nt) in the RNAseq libraries, so we do not know whether these stable introns exist as lariats (with the 3' tail) or 2'−5' circles (without the 3' tail).

We asked if these 14 introns had any other unusual sequence features. We derived sequence logos centered at their aligned 5' splice sites, branchpoints, and 3' splice sites. Their 5' and 3' splice sites conform to the standard consensus sequences, but their branchpoint follows a CC consensus 17–49 nucleotides upstream of 3' splice site (*Figure 6B*; *Figure 6—source data 1*). Mismatches in read alignments at the branchpoint and the relative scarcity of circularly permuted reads versus linear reads were consistent with the existence of a 2'−5' linkage that the reverse transcriptase traverses with lower efficiency and accuracy. Other than conservation of 5' and 3' splice site sequences, none of the 14 free introns showed notable phylogenetic sequence conservation (*Figure 6C*). Although we hypothesize that the C branch site is an essential feature in stabilizing these projection-localized free introns, C branchpoints were not notably conserved across multi-species alignments of these introns, indicating little evolutionary pressure to conserve this feature.

## Discussion

In this study, we conducted a comprehensive analysis of transcripts localizing to neuro-glial projections of primary rat hippocampal cells. Our data add to the growing compendium of localized RNAs identified using high-throughput methods in diverse rat/mouse neuronal cell types (e.g., motor neurons, *Briese et al., 2016*; dorsal root ganglia, *Gumy et al., 2011*, *Minis et al., 2014*; retinal ganglion cells, *Zivraj et al., 2010*; neuropil, *Cajigas et al., 2012*; cortical cells, *Taliaferro et al., 2016*, *Taylor et al., 2009*, *Poulopoulos et al., 2019*; primary hippocampal cells, *Poon et al., 2006*, *Miyashiro et al., 1994*). Because our main intent was to identify and characterize intron sequences in projections, we created rRNA-depleted total RNAseq libraries from rat hippocampal cells grown on membranes that provided physical separation between projections and cell bodies (*Figure 1*). As expected, 97% of the annotated intron regions that we could detect across all samples met our expression cutoffs in whole cells only (*Figure 3*). Of the 3% of intron regions that are also detectable in projections, the majority turned out to be attributable to incomplete annotation of alternative

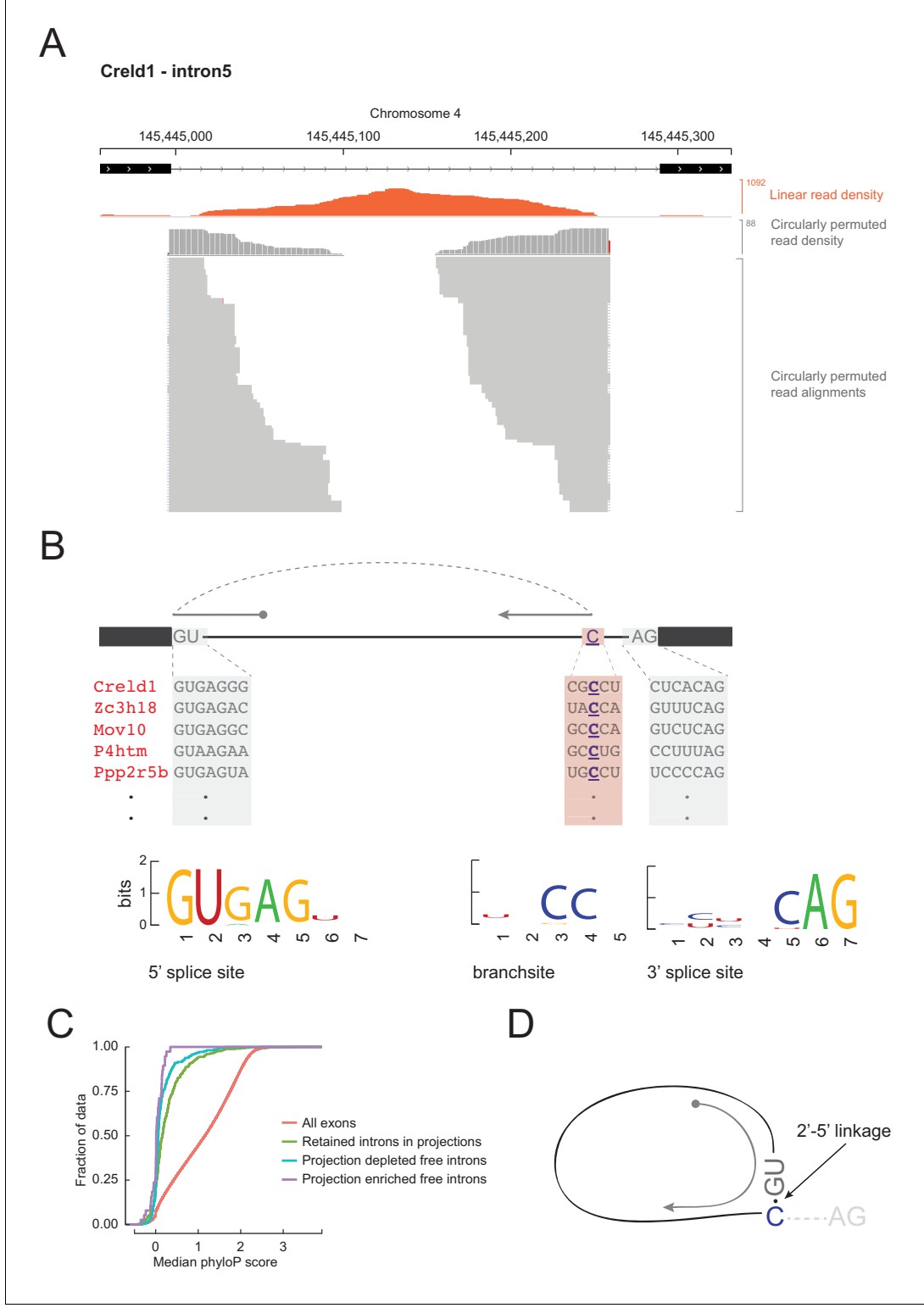

**Figure 6.** Free circular introns with a noncanonical branchpoint. (**A**) Projection RNAseq read density for linearly mapped (orange) and circularly permuted (gray) reads across Creld1 - intron5. Alignment mismatches at the junction of circularly permuted reads are colored. Individual circularly permuted read alignments are shown as horizontal gray bars, with each bar representing one read. (**B**) Schematic of circularly permuted read alignment and sequence composition at 5' splice sites, branch sites (the branchpoint nucleotide is underlined), and 3' splice sites of five example introns. (**C**) Cumulative distribution function plots for phyloP sequence conservation showing

*Figure 6 continued on next page*

*Figure 6 continued*

lack of conservation in projection-enriched free introns relative to exons, or even to retained introns. *x*-axis represents median phyloP score over a 50 nt sliding window with 10 nt step size. (**D**) Inferred molecular structure of projection-localized free circular introns.

The online version of this article includes the following source data for figure 6:

**Source data 1.** Characteristics of circularly permuted read alignments on free introns.

mRNA isoforms or to intron-encoded snoRNAs/scaRNAs (*Figure 4*). Integration of layers of information – polyA+ RNAseq, ribosome profiling, and PASseq libraries – from parallel hippocampal cultures proved essential for us to characterize these regions.

Full-length introns present at high abundance in projections fell into two classes, one expected (retained introns) and one unexpected (free introns). As a class, retained introns displayed no strong tendency toward preferential projection localization (*Figure 4—figure supplement 2*). Thus, while some transcripts may contain localization elements in an alternatively retained intron (*Sharangdhar et al., 2017*), this does not appear to be a general function of intron retention in neurons. Some retained introns clearly express alternative protein isoforms, as we observed for *Sept3* intron 10 (*Figure 4*). Notably, *Kcnma1* intron 23, previously proposed to undergo local splicing in primary rat hippocampal dendrites (*Bell et al., 2010*), had almost no coverage in our projection libraries in the same cell types (*Figure 4—figure supplement 3*). Further, the retained introns that we identified in neuro-glial projections exhibit almost no overlap with the set of 'cytoplasmic intron sequence-retaining transcripts' (CIRTs) previously reported to localize to rat primary hippocampal dendrites (*Figure 3—figure supplement 2*) (*Buckley et al., 2011*). CIRTs and *Kcnma1* intron 23 were initially identified by sequencing RNA from 15 to 300 individually dissected dendrites. While the method was state-of-the-art at the time, the low amount of input RNA necessitates multiple rounds of RNA and cDNA amplification prior to sequencing. In contrast, we started with high RNA input, restricted cDNA amplification to ≤ 15 cycles, and sequenced deeply to capture even low abundance RNA species.

Our most surprising finding was the detection of free introns in projections (*Figure 5*), which we infer to be lariat species containing a cytosine or guanine branchpoint instead of the canonical adenosine (*Figure 6*). The most likely explanation for the persistence of such species is the inability of debranching enzyme to cleave the 2′−5′ bond at a C or G branch. That stable circular introns (or stable intron sequences, sisRNAs) can escape the nucleus and accumulate in the cytoplasm has already been reported by *Talhouarne and Gall (2018)*. In that study, the authors observed sisRNAs across diverse samples, including cell lines from multiple vertebrate species and mouse fibroblasts, red blood cells, liver, and brain. Despite this pervasiveness, these introns' general lack of sequence conservation and absence of interspecies overlap with sisRNAs strongly disfavors any evolutionarily conserved function. Rather, it seems more likely that sisRNAs are gene expression byproducts, perhaps with little positive or negative influence on cellular function; they may simply be noise in the system. We hypothesize that substituting any intron branchpoint from a canonical A to C/G has little effect on splicing while making the excised intron resistant to lariat debranching, leading to a stable 2′−5′ circular RNA without much phenotypic consequence, and that such substitutions are tolerated at low frequency in any organism with spliceosomal introns in its genes.

Our study extends previous work on sisRNAs by the Gall lab by finding that sisRNAs are enriched in neuro-glial projections. If these free circular introns lack evolutionary signatures of function, why would they appear to be enriched in projections? We observe that because RNAseq experiments measure relative rather than absolute abundances, relative enrichment does not necessarily imply active localization. We hypothesize that mRNAs are typically occupied by polyribosomes in the cell body, whereas an untethered stable noncoding RNA may more freely diffuse throughout the entire volume of the cell, including distant projections. Unlike most mRNAs, free circular introns showed no evidence of ribosome occupancy. It has previously been observed for *Actb* mRNA in neurons that a lack of ribosomal engagement leads to faster mRNA diffusion kinetics than is observed for actively translating mRNAs (*Katz et al., 2016*). The intracellular environment of neurons and neuronal projections is full of highly motile proteins and organelles that are themselves actively transported (*Stępkowski et al., 2017*), as indeed we see reflected by the prevalence of mitochondrial RNA in our projection libraries (> half of all reads in our projection libraries were of mitochondrial origin;

*Figure 1—figure supplement 1*). We imagine that the movement of large objects within the confines of narrow cellular projections mixes the surrounding cytoplasm and could well force unanchored molecules to move in both anterograde and retrograde directions. Thus, free stable circular introns may appear to be *relatively* enriched in projections simply because they are long-lived and more freely diffusing than other RNAs that are depleted in projections because they are less stable or tethered in the cell body (to polyribosomes, for example).

This hypothesis has implications for another long-standing puzzle in neuronal RNA localization. All transcriptome-wide studies of the mRNA content of dendrites, axons, or synapses, including ours, have observed an enrichment of the complete set of ribosomal protein (RP) mRNAs (*Moccia et al., 2003*; *Poon et al., 2006*; *Zivraj et al., 2010*; *Cajigas et al., 2012*; *Puthanveettil et al., 2013*; *Ainsley et al., 2014*; *Briese et al., 2016*; *Taliaferro et al., 2016*; *Nakayama et al., 2017*; *Shigeoka et al., 2018*, to cite a few). Ribosomes are assembled in the nucleus, so localization of RP mRNAs to distant projections is puzzling. Recently, ribosome protein synthesis and incorporation of some individual ribosomal proteins into assembled ribosomes in axons of Xenopus retinal cells has been shown to occur and to be important for axon development (*Shigeoka et al., 2018*), however, only a subset of RP mRNAs in axons are translated. It has been previously shown that in quiescent or growth-arrested cells (such as neurons), 30–40% of RP mRNA molecules sediment in polysome profiles as free mRNPs, compared to only ~10% for non-RP mRNAs (*Meyuhas et al., 1987*). Thus, it appears that in neurons, a substantial fraction of RP mRNAs are not engaged with ribosomes, and like our free circular introns, unengaged RP mRNAs may be more free to diffuse about the cell. We propose that the relative enrichment of RP mRNAs in projections could simply be due to their low ribosome engagement, not to an active transport mechanism.

## Materials and methods

### Primary hippocampal neuron culture

Animals were handled in accordance with protocols approved by the Institutional Animal Care and Use Committee at University of Massachusetts Medical School and Harvard University. All experiments were performed on primary hippocampal neurons of embryonic day 19 rat fetuses. Pregnant Sprague Dawley rats purchased from Charles River Laboratories at 19 days of gestation were euthanized by carbon dioxide asphyxiation immediately followed by diaphragm puncture to ensure death of the animal prior to surgical removal of fetuses. Fetuses were transferred to pre-cooled dishes and placed on ice. Fetal brains were gently extracted under sterile conditions and submerged in ice-cold sterile HBSS (Hank's Balanced Salt Solution, Gibco 14185052) for transport to the tissue-culture laboratory. The brains were transferred to ice-cold Hibernate E (BrainBits) for microdissection of the hippocampi in a sterile tissue culture hood. Isolated hippocampi were transferred to freshly prepared, pre-warmed pre-dissociation solution (comprising Hibernate E, EBSS, Papain and DNase from Worthington LK003176) and incubated at 37°C for 30 min. Pre-dissociation solution was replaced by 2 ml MEM complete media (containing 50 ml 10x MEM (Invitrogen 11430–030), 15 ml 20% glucose, 15 ml 7.5% sodium bicarbonate, 2 ml 1N HCl, 400 ml water, 5 ml 200 mM glutamax (Gibco 35050061), 50 ml heat inactivated horse serum, 5 ml penicillin-streptomycin (Gibco 15140122)) and the hippocampi were dissociated by gentle trituration, first with a regular and then a fire-polished (with reduced tip diameter) glass pipette. Additional MEM complete media was added to the dissociated cells. Cells were counted using a hemocytometer and plated at optimized densities on poly-D-lysine hydrobromide (Sigma #P0899) coated plates/surfaces. Two hours after plating, MEM complete media was replaced with Neurobasal media (500 ml Neurobasal (Invitrogen 21102–049), 1.3 mL 200 mM glutamax, 1X antibiotic/antimycotic (Gibco 15240062), 1x B27 (Gibco 17504001)). We obtained ~13 embryos per rat and ~1 million cells per embryo.

Mature neurons are post-mitotic but glial cells divide. To prevent glial cells from crowding out the neurons, we treated the culture with a DNA replication inhibitor, Cytosine β-D-arabinofuranoside (Sigma-Aldrich #C1768), on the third day in vitro (DIV). Half of the neurobasal media was replaced with fresh media on DIV = 6, and then on DIV = 12. Cells were harvested on DIV = 14.

## Physical separation of neuro-glial projections from cell bodies

To separate neuro-glial projections from cell bodies, primary neurons were cultured on Falcon Permeable Support with Polyethylene Terephthalate (PET) membranes (Corning Life Sciences #C353102) that have 1 µm pores such that the cell bodies remain on top of the membrane, but cellular projections (axons, dendrites, and glial projections) can grow through the pores to the underside of the membrane (*Poon et al., 2006*). Lysate collected from the top surface of the membranes comprises whole cells (cell bodies and projections) whereas lysate from the underside of the membranes comprises neuro-glial projections and some cytoplasmic blebs.

Primary hippocampal neurons were cultured at a density of $0.2 \times 10^6$ per well on Falcon Permeable Support designed to fit 6-well tissue culture plates (Corning Life Sciences #C353102). The plating density was optimized to ensure healthy neuronal cultures for harvesting on DIV = 14. Before setting up the neuronal culture, membranes were immersed in poly-D-lysine hydrobromide solution (0.1 mg/ml in 0.05 M sodium borate pH 8.5) overnight at 37°C. Before plating neurons, membranes were rinsed three times with sterile water and incubated at 37°C for at least 2 hr while immersed in MEM complete media. Neuronal health was assessed for every biological replicate by immunofluorescence imaging using a dendritic marker, MAP2, to visually inspect cell morphology.

## RNA isolation from neurons cultured on semipermeable membranes

To extract RNA from neuro-glial projections, we removed media from the Falcon Permeable Support, turned it upside-down with the projections-side of the membrane facing upward, applied 200 µl of TRIzol reagent (ThermoFisher #15596018) to the membrane, quickly scraped the surface with one stroke using a cell-scraper, tilted the membrane and gently collected the lysate from the edge of the membrane with a pipette. Lysate was similarly collected from the cell body side of the membrane by applying 500 µl of TRIzol reagent.

RNA was extracted following the steps recommended by TRIzol reagent manual with minor modifications. Briefly, after application of TRIzol reagent, the lysate was transferred to 15 ml Falcon tubes and vortexed for 30 s, followed by a 5 min incubation at room temperature to dissociate nucleoprotein complexes. 200 µl of chloroform per 1 ml of TRIzol was then added for phase separation. The tubes were vigorously shaken by hand and centrifuged at 12,000 x g for 15 min at 4°C. The aqueous phase was transferred to fresh tubes and the chloroform wash was repeated two more times. RNA was precipitated by mixing with 100% isopropanol, incubating at room temperature for 15 min, and centrifugation at 12,000 x g at 4°C for 15 min. The supernatant was removed without disturbing the precipitated RNA. The RNA precipitate was rinsed two times with 75% ethanol, air dried, and dissolved in RNase-free water by incubating for 10 min at 55–60°C. RNA was stored at −80°C until needed.

Typical RNA yield from neuronal cultures on $8 \times 6$ well plates with the semipermeable membrane inserts was ~5 µg from the projections lysate for every ~50 µg from the whole cell lysate. The quality and quantity of RNA was assessed by Nanodrop UV spectrophotometer (A260/A280 measurements) and Bioanalyzer Pico RNA microcapillary electrophoresis (*Figure 1—figure supplement 2*).

## rRNA-depleted total RNAseq library preparation

RNAseq libraries were prepared following the protocol published in *Zhang et al. (2012)*. Briefly, 5 µg of RNA from each sample was treated with TURBO DNase (ThermoFisher AM2238) followed by clean-up and enrichment of RNA > ~150 nt using RNA clean and concentrator (Zymo Research #R1013). DNase activity was tested beforehand to ensure DNA digestion. Total RNA was depleted of rRNA using the Ribo-Zero rRNA removal kit (Illumina #MRZH11124), following manufacturer's protocol. RNA was hydrolyzed using 5X first strand buffer (provided with Superscript III reverse transcriptase, ThermoFisher #18080044) at 94°C for 4 min and 50 s and immediately moved to ice. The fragmented RNA was reverse transcribed using random hexamers (ThermoFisher #N8080127) and Superscript III to make single-stranded cDNA. To make strand specific libraries, the second strand complementary to the cDNA was transcribed with DNA polymerase I (New England Biolabs #M0209S) using dUTP instead of dTTP. Illumina sequencing adapters were ligated to the double stranded cDNA. The dUTP-containing strand was degraded using Uracil-DNA Glycosylase (New England Biolabs #M0280S). The resulting single-stranded cDNA was amplified with 13 or 15 PCR cycles followed by size-selection using Pippin Prep (Sage Science) to select for reads with cDNA

inserts of 150–450 nt length. Before subjecting to Illumina sequencing, the RNAseq library quality was assessed by running the samples on the Bioanalyzer and Sanger sequencing a subset of TOPO-TA cloned products. Only those libraries with sufficient final concentration (>12 nM), a product of appropriate size range (mode 300 nt size), and comprising expected RNA sequences (for instance, exon regions of abundant mRNAs) were selected for Illumina sequencing.

A total of five biological replicates were sequenced over a span of 2 years, replicates 1–3 (paired-end 100) in 2014 and replicates 4,5 (paired-end 125) in 2016.

## PolyA site sequencing

PolyA site sequencing (PASseq) libraries were prepared following the protocol in *Ashar-Patel et al. (2017)* and *Heyer et al. (2015)*. 2–5 µg of total RNA from each sample (three biological replicates of projection and whole-cell lysates) was treated with DNase and fragmented as described above. The RNA was reverse transcribed with Superscript III using an anchored oligo-dT primer containing Illumina sequencing adapters and a unique barcode for each sample. Single-stranded RNA was degraded with RNaseI. The cDNA was denatured (65°C for 5 min) and resolved by electrophoresis on a 10% polyacrylamide gel to select 160–210 nt sized fragments (for a 50–100 nt expected insert size sans the adapter sequences). To extract cDNA from the gel, a piece of the gel containing the cDNA was cut at the appropriate location, crushed, and nutated overnight in a solution of 300 mM sodium chloride and 10 mM EDTA. The solution was recovered from gel pieces by centrifugation in Corning Costar Spin-X columns (#07200386) at 10,000 x g for 3 min. cDNA was precipitated using isopropanol, followed by washes in 70% ethanol. The cDNA was then circularized using CircLigase (EpiCentre BioTechnologies #CL4115K) and amplified with 12–14 PCR cycles. The amplified DNA library was further enriched for a product of size 180–280 nt to exclude insert-less product (150 nt) using Pippin Prep.

## Ribosome profiling and polyA+ RNAseq from the cytoplasmic fraction

Ribosome profiling and corresponding RNAseq libraries were prepared from fractionated cytoplasmic lysate of primary hippocampal neurons following the protocol in *Ricci et al. (2014)* and *Heyer et al. (2015)*.

Rat primary hippocampal neurons were cultured (as described above) on poly-D-lysine coated 6 cm plates (three plates per sample) at a density of $1 \times 10^6$. After 14 days in vitro, cyclohexamide was added to the media at 100 µg/µl final concentration for 10 min to stall translation. The plates were placed on ice where the media was removed and the cells were washed two times with 2 ml ice-cold PBS containing 100 µg/ml cyclohexamide. The cells were lysed in 200 µl lysis buffer (10 mM Tris-HCl pH 7.5, 5 mM $MgCl_2$, 100 mM KCl, 1% Triton X-100, 2 mM DTT, 100 µg/ml cycloheximide, protease inhibitor (Complete, EDTA-free, Roche)). The lysate was collected by scraping the plates, transferred to a clean microcentrifuge tube and incubated on ice for 5 min, followed by centrifugation at 1300 x g for 10 min to pellet the nuclei. The supernatant was recovered, flash frozen in liquid nitrogen, and stored at −80°C until needed. Half of the lysate was used for ribosome profiling and the other half for polyA+ RNAseq library preparation.

To purify ribosome occupied RNA sequences for ribosome profiling, the RNA was digested with 300 units of RNase T1 (Fermentas) and 500 ng of RNase A (Ambion) for 30 min at room temperature to break down polysomes into monosomes. The monosomes were purified by density gradient ultracentrifugation. Lysates were fractionated by centrifugation through a 10–50% (weight/volume) linear sucrose gradient (20 mM HEPES-KOH, pH 7.4, 5 mM $MgCl_2$, 100 mM KCl, 2 mM DTT, 100 µg/ml cyclohexamide) at 35,000 r.p.m. for 2 hr and 40 min at 4°C. A gradient fractionator (Brandel) was used to identify and collect the monosome enriched fraction by measuring absorbance at 254 nm. RNA was extracted from the monosome fraction and resolved by electrophoresis on a denaturing polyacrylamide gel to select RNA fragments ranging from 26 to 32 nt in size.

To extract RNA for ribosome profiling and polyA+ RNAseq, SDS was added to 1% final volume, and proteinase K (Invitrogen) was added to a final concentration of 200 µg/ml. The samples were incubated at 42°C for 45 min. One volume of acid phenol/chlorofom (Ambion AM9720, pH 4.5) was added and the samples were vortexed for 30 s followed by centrifugation at 12,000 x g for 15 min. The supernatant was transferred to a clean microcentrifuge tube and 0.1 vol of sodium acetate (3 M, pH 5.2) and 10 mM final concentration of $MgCl_2$ were added. To precipitate RNA, 1 vol of 100%

isopropanol was added to the solution and centrifuged at 12,000 x g for 35 min. The precipitated RNA was rinsed with 70% ethanol, air dried, and reconstituted in 5 µl water.

For polyA+ RNAseq library preparation, the RNA was partially hydrolyzed using Fragmentation Reagent (Ambion) prior to cDNA library preparation.

To prepare cDNA libraries from ribosome occupied RNA fragments and from RNA fragments from the cytoplasmic lysate, the 3' ends of RNA fragments were dephosphorylated with T4 polynucleotide kinase (New England BioLabs #M0201S). A preadenylated DNA adaptor sequence was ligated to the 3'-hydroxyl ends of the RNA fragments using T4 RNA Ligase (T4 RNA Ligase 2, truncated K227Q, NEB #M0351S). The ligated RNA product was reverse transcribed using Superscript III and a barcoded primer with sequence complementarity to the adaptor. The reverse transcription primer also contained adaptors required by Illumina sequencers. The resulting cDNAs were enriched for desired product size, circularized, and amplified following the steps described for PASseq library preparation.

Ribosome profiling libraries were amplified with 8 PCR cycles whereas polyA+ RNAseq libraries were amplified with 13 PCR cycles and sequenced on Illumina HiSeq sequencers for 50 nt single reads.

## Immunofluorescence staining

Cells were cultured either on semipermeable membrane inserts or Thermo Scientific Lab-Tek II chamber slides as described in previous sections. At DIV = 14, media was removed, the cells were rinsed two times with phosphate buffered saline (PBS, pH = 7.4) and subsequently treated with fixative (4% paraformaldehyde) for 10 min at room temperature. The fixative was removed and the cells were rinsed three times with PBS followed by permeabilization with 0.1% Triton X-100 for 5 min. Cells were then rinsed three times with PBS and quenched with 50 mM ammonium chloride (in double-distilled water) for 10–15 min, followed by three more rinses with PBS. The fixed and permeabilized cells were then incubated with 10% normal goat serum in PBS (blocking solution) for 30 min at room temperature. For cells grown on membranes, the membranes were cut out of their plastic support system using a sharp blade and transferred to small chambers for the next steps. The samples were kept moist at all times during the protocol.

The membranes or slides were incubated overnight at 4°C with primary antibody diluted in blocking solution then rinsed three times with PBS. Fluorescently-labeled secondary antibody was applied for 1 hr in the dark at room temperature and rinsed by three washes with PBS. The membranes were cut in half and placed on glass slides with either the whole cell or the projection side on top. Pro-Long Gold antifade media with DAPI was applied to the membranes before covering them with a 0.16–0.19 mm thick cover glass. The edges of cover glass were sealed with transparent nail-polish and allowed to set overnight in the dark at room temperature. The samples were imaged on DeltaVision or Zeiss Cell Discoverer microscopes.

## smFISH

smFISH was performed following ACDBio protocol. Briefly, cells were cultured on Ibidi poly-D-lysine coated chambered coverslips. At DIV = 14, media was removed, cells were rinsed two times with PBS, fixed for 30 min at room temperature, then rinsed three times with PBS. The cells were dehydrated by incubating them in sequentially higher concentrations of ethanol (50%, 70%, and 100%, respectively) for 5 min each and a final immersion in 100% ethanol for at least 10 min at room temperature. The dehydration was then reversed by incubation in 70% ethanol for 2 min, 50% ethanol for 10 min, and rinsing with PBS at room temperature. They were then treated with Protease III (ACD Bio) for 10 min at room temperature. smFISH probes were hybridized following manufacturer's instructions. In the end, the cells were counterstained with DAPI and mounted with ProLong Glass Antifade Mountant (P36980). Probes were multiplexed to image up to three different targets in the same sample. The samples with imaged on Zeiss Cell Discoverer at the Harvard Center for Biological Imaging. Regions targeted by smFISH probes are listed in *Figure 2—source data 2*.

## Computational analysis

An outline for the data analysis workflow is shown in *Figure 1—figure supplement 3* and *Figure 3—figure supplement 1*. Code and intermediate data files for reproducing and extending these

analyses are available as a tar archive at http://eddylab.org/publications/Saini19/Saini19-supplement.tar.gz. Descriptive placeholder input and output filenames (enclosed in <>) are used in the commands shown. If reusing these commands, please replace placeholder filenames with appropriate ones and omit enclosing '<' and '>'.

## rRNA-depleted, total RNAseq genome alignment

Paired-end reads from five biological replicates (10 samples total) of projection and whole cell RNA-seq are provided in the fastq format at GSE129924. Reads from replicates 1–3 are 2 × 100 nt, whereas replicates 4 and 5 are 2 × 125 nt long. Replicates 4 and 5 also contain ERCC RNA spike-in. Reads from all 10 samples were aligned to ERCC, rRNA, repeat elements cataloged by Repeat-Masker (*Jurka, 2000*), 7SL or SRP (the RNA component of signal recognition particle), and the mitochondrial genome, serially in that order, using bowtie2 version 2.2.3 (*Langmead and Salzberg, 2012*). Unaligned reads after each step were passed on for the next alignment. The following parameters were used for each alignment:

```
$ bowtie2-2.2.3/bowtie2 -p 2 -N 1 –no-unal \\
--un-conc <unaligned.fastq> --al-conc <aligned.fastq> \\
-1 <read1.fastq> -2 <read2.fastq> -S <alignments.sam>
```

Unaligned or filtered reads were then mapped to the rat genome (Ensembl release 81, Rnor_6.0, annotation downloaded on July 24, 2015) (*Zerbino et al., 2018*) with TopHat version 2.1.1 (*Kim et al., 2013*) using the following parameters:

```
$ tophat/2.1.1/tophat -p 6 --library-type fr-firststrand \\
--b2-sensitive --mate-inner-dist 100 -i 30 -g 10 \\
--max-coverage-intron 5000000 -G <genes.gtf> -o <output_directory> \\
<genome_index_base> <filtered_read1.fastq> <filtered_read2.fastq>
```

High quality read alignments were selected (using SAMtools version 1.4.1) for visualization on the rat genome browser.

```
$ samtools view -bh -q 10 <alignment.bam> > <alignment.q10.bam>
```

## PolyA site identification from PASseq data

Six samples (three from whole cell and three from projection lysates) were barcoded and sequenced in one lane on NextSeq 500 with 150 cycles. The barcoded reads were parsed using Illumina's bcl2fastq version 1.8.4 conversion software.

```
$ bcl2fastq --barcode-mismatches 1 -R <run_directory> \\
-o <output_directory> --use-bases-mask I5y*n 2 > <output.log>
```

Reads were trimmed using cutadapt version 1.7.1 (*Martin, 2011*) to remove stretches of A's from the 3' end and barcode sequence from the 5' end, and then selected for minimum 25 nt resulting read length. We used the fastx toolkit version 0.0.14 (http://hannonlab.cshl.edu/fastx_toolkit/) to further select reads with sequencing quality greater than 35. Parsed, trimmed, and filtered fastq files are provided on GSE129924.

```
$ cutadapt -a AAAAAAAA -o <A_trimmed.fastq> <input.fastq> \\
2 > <output.log>
$ cutadapt -u 7 -m 25 -o <A_barcode_trimmed.fastq> \\
<A_trimmed.fastq> 2 > <output.log>
$ fastq_quality_filter -v -q 35 -p 50 -i <A_barcode_trimmed.fastq> \\
-o <A_barcode_trimmed_hiqual.fastq> 2 > <output.log>
```

For PAS site identification, we used cleanUpdTSeq v.1.0.2 (*Sheppard et al., 2013*), which calculates the probability of a genomic locus to be a true polyadenylation site. The reads were first aligned to the rat genome using TopHat version 2.1.1 (*Langmead and Salzberg, 2012*), then reads with mapping quality greater than 10 were selected using SAMtools version 1.4.1 (*Li et al., 2009*).

```
$ tophat/2.1.1/tophat -p 4 --library-type fr-secondstrand \\
--b2-very-sensitive --no-novel-juncs -i 30 -g 10 \\
-G <genes.gtf > -o <output_directory>\\
<genome_index_base> <PASseq_reads.fastq>
$ samtools view -bh -q 10 <alignment.bam> > <alignment.q10.bam>
```

Strand-specific 3' end read alignment coordinates were extracted using BEDTools version 2.26.0 (*Quinlan and Hall, 2010*).

```
$ bedtools genomecov -3 -d -strand - -ibam <alignment.q10.bam> \\
-g <rnor6.ChromInfo.txt> > <output.q10.3cov.minus.txt>
$ bedtools genomecov -3 -d -strand + -ibam <alignment.q10.bam> \\
-g <rnor6.ChromInfo.txt> > <output.q10.3cov.plus.txt>
```

Only those genomic loci with more than five read alignments were considered for further analyses.

```
$ awk '{if ($3 > 5) print $0}' <*.3cov.minus.txt> \\
> <*.3cov.minus.non0.txt>
$ awk '{if ($3 > 5) print $0}' <*.3cov.plus.txt> \\
> <*.3cov.plus.non0.txt>
```

To prepare data for use in CleanUpdTseq, files were converted to bed format and a unique identifier for each site was added. Each line of the bed file contained single nucleotide genomic coordinates and the number of reads whose 3' ends aligned at that locus.

```
$ awk '{OFS = "\t"}; {print $1 , $2 , $2 , "SampleID_"NR, $3 , "+"}' \\
> <3cov.plus.bed>
$ awk '{OFS = "\t"}; {print $1 , $2 , $2 , "SampleID_"NR, $3 , "-"}' \\
< <3cov.minus.bed>
```

Bed files from the plus and minus strands were combined and sorted, and then they were split by chromosome.

```
$ cat <3cov.plus.bed> <3cov.minus.bed> | sort -k 1,1 -k 2,2n \\
> <output.bed>
$ awk '{if ($1 == "chr1") print $0} ' <*.bed> > <*.chr1.bed>
$ awk '{if ($1 == "chr2") print $0} ' <*.bed> > <*.chr2.bed>
...
$ awk '{if ($1 == "chrY") print $0} ' <*.bed> > <*.chrY.bed>
```

polyA sites were then identified using cleanUpdTSeq (*Sheppard et al., 2013*). Genomic loci with a high probability of being a true polyadenylation site (p-value < 0.001) were selected for further analysis or visualization on the rat genome browser.

## Cytoplasmic polyA+ RNAseq and ribosome profiling genome alignment

The adapter was trimmed from fastq files, and only reads longer than 24 nt were kept. Reads were filtered for rRNA, tRNA, repeat elements, and mitochondrial genome following the steps described in the PASseq methods section. Reads were aligned to the genome with TopHat using RefSeq annotation (RGSC 6.0/rn6, Jul. 2014, downloaded on July 7, 2016) as a reference. The alignment is reported in bam file format.

```
$ tophat/2.0.14/tophat -p 4 --library-type fr-secondstrand \\
--b2-sensitive -g 10 --keep-tmp -G <genes.gtf> \\
-o <output_directory> <genome_index_base> <filtered_reads.fastq>
```

## Differential expression analysis

To quantify annotated transcript abundance, we used Kallisto version 0.44.0 (*Bray et al., 2016*) with the following parameters:

```
$ kallisto quant -i genes.fa.idx -o ./kallisto_output \\
-b 100 --rf-stranded <read1.fastq.gz> <read2.fastq.gz>
```

Reference sequences (fasta format) of protein coding and noncoding RNAs were downloaded from ensembl.org (*Zerbino et al., 2018*) in January, 2018. Transcripts from the mitochondrial genome were omitted from this analysis. The sequence of nuclear noncoding RNA, *Xist*, was imported from RefSeq (NR_132635.1) (*O'Leary et al., 2016*) because it was not annotated in the rat genome reference data downloaded on July 24, 2015 (Ensembl release 81, Rnor_6.0).

To identify RNAs enriched in projections, we compared transcript abundances in projections to whole cells using Sleuth version 0.30.0 (*Pimentel et al., 2017*). The output is provided in *Figure 2—source data 1*.

## Gene ontology analysis

To identify gene families enriched in projections or whole cells, we used GeneCodis http://genecodis.cnb.csic.es/ (accessed on November 5, 2018) (*Carmona-Saez et al., 2007*) with default parameters, focusing on the cellular component for gene ontology classification. RNAs enriched in projections ($n$ = 1,440) or whole cells ($n$ = 1,297) were selected using a q-value cutoff of < 0.01 and $\log_e$ mean read counts across all libraries >1 and compared against a background comprising all RNAs that had $\log_e$ mean read counts > 1 ($n$ = 19,461).

## Intron quantification

To define intron regions, we first extracted the genomic coordinates of all annotated exons in any isoform of a gene as a .gff file using dexseq_prepare_annotation.py from DEXSeq (*Anders et al., 2012*). Intron regions were defined as the non-exon regions of each gene. Intron region coordinates were derived from exons using a program written by Alejandro Reyes (ORCID: 0000-0001-8717-66) copied from http://seqanswers.com/forums/showpost.php?p=137918&postcount=4.

To count the number of reads aligning to intron regions, exon-intron (EI) boundaries, intron-exon (IE) boundaries, and exon-exon (EE) junctions, we used BEDTools version 2.26.0 (*Quinlan and Hall, 2010*). Reads were aligned to the genome using TopHat, a splice-sensitive alignment algorithm, as described above. Only reads with high quality alignments were counted on regions of interest.

The intron reference file was first converted from .gff to .bed file format using BEDOPS (*Neph et al., 2012*). EI and IE regions span 50 nt, 25 nt from the exon and 25 nt from the intron. The EE region coordinates include the intron region and 50 nt of each flanking exon. To count reads crossing the EE junction, only those read alignments that started and ended at the EE coordinates were considered. Since EE regions are longer than 100 nt because they include the intron, the only way a 100 nt long read would start and end at the EE coordinates is if the alignment splits. In case of ribosome profiling and polyA+ RNAseq data, where the minimum read length is 24 nt, the EE regions included the intron region and 12 nt of each flanking exon. The commands used to align the reads to the rat genome, define regions of interest, and count reads on them are shown in *Figure 1—figure supplement 3* and *Figure 3—figure supplement 1*.

## Circularly permuted read alignments

To align reads allowing circular permutation, we used find_circ.py version 1.2 (*Memczak et al., 2013*) following instructions provided on the github repository https://github.com/marvin-jens/find_circ. Reads that failed to align to the rat genome using TopHat (see above) were used to search for circularly permuted alignments. The following commands were executed:

```
$ unmapped2anchors.py <unmappedreads.bam> > <unmapped_anchors.qfa>

$ bowtie2 -p 8 --score-min=C,-15,0 --reorder –mm -q \\
-x <Bowtie2Index/genome> -U <unmapped_anchors.qfa> | \\
find_circ.py -G <genome.fa> -B <anchors.bam> –noncanonical \\
-R <spliced_reads.txt> -s <stats.txt> > <circs.txt>
```

### Sequence conservation

To assess sequence conservation of regions of interest (free intron regions enriched in and depleted from projections, retained introns in projections, and all annotated exons), we used the PhyloP sequence conservation scores from 20 aligned vertebrate genomes (*Pollard et al., 2010*) downloaded from the UCSC database (*Kent et al., 2002*). Each nucleotide has a PhyloP conservation score. For every region of interest, we calculated the average PhyloP score across 50 nt windows with a 10 nt interval (using bigWigAverageOverBed version two from *Kent et al., 2010*) and took the median score across all windows in the region.

```
$ bedtools makewindows -w 50 -s 10 -b <region.bed> \\
-i srcwinnum > <windows.bed>

$ bigWigAverageOverBed –bedOut=<avg_phyloscores.bed> \\
<rn6.phyloP20way.bw> <windows.bed> <avg_phyloscores.tab>
```

## Acknowledgements

We thank Guiping Wang and Xiaowei Zhuang (Harvard University) for providing reagents and technical assistance with smFISH experiments; the deep sequencing core facility at University of Massachusetts Medical School (UMass) for HiSeq sequencing; Massachusetts Green High Performance Computing Center (UMass) for computational resources; Harvard Center for Biological Imaging for access to microscopes and image processing resources; Harvard MCB graphics for feedback on data presentation; Phillip Zamore (UMass) and members of the Eddy lab (Harvard University) for discussions and critical reading of the manuscript. This work was supported by funding from HHMI (SRE), NIH (R01-GM53007 to MJM), and HHMI International Student Research Fellowship (HS). MJM was an HHMI Investigator at the time this study was conducted.

## Additional information

### Funding

| Funder | Grant reference number | Author |
| --- | --- | --- |
| National Institute of General Medical Sciences | R01-GM53007 | Melissa J Moore |
| Howard Hughes Medical Institute | | Sean R Eddy Melissa J Moore |
| Howard Hughes Medical Institute | International Student Research Fellowship | Harleen Saini |

The funders had no role in study design, data collection and interpretation, or the decision to submit the work for publication.

### Author contributions

Harleen Saini, Conceptualization, Resources, Data curation, Software, Formal analysis, Validation, Investigation, Visualization, Methodology, Writing—original draft, Writing—review and editing; Alicia A Bicknell, Resources, Methodology, Writing—review and editing; Sean R Eddy, Melissa J Moore,

Conceptualization, Resources, Supervision, Funding acquisition, Methodology, Writing—original draft, Project administration, Writing—review and editing

### Author ORCIDs
Harleen Saini (iD) https://orcid.org/0000-0001-6954-4098
Sean R Eddy (iD) https://orcid.org/0000-0001-6676-4706

### Ethics

Animal experimentation: This study was performed in strict accordance with the recommendations in the Guide for the Care and Use of Laboratory Animals of the National Institutes of Health. Animals were handled in accordance with protocols approved by the Institutional Animal Care and Use Committees at University of Massachusetts Medical School (docket #A-2245-16) and Harvard University (protocol #10-16-1). All surgery was performed after euthanization, and every effort was made to minimize suffering.

### Decision letter and Author response
Decision letter https://doi.org/10.7554/eLife.47809.sa1
Author response https://doi.org/10.7554/eLife.47809.sa2

## Additional files

### Supplementary files
• Source code 1. A tarball of supplementary tables and code.

• Transparent reporting form

### Data availability

All raw data (fastq format) and corresponding coverage files (bigwig format) are available at NCBI GEO under accession number GSE129924, and a tarball of supplementary tables and code has been uploaded as Source code 1 and is also available at http://eddylab.org/publications/Saini19/Saini19-supplement.tar.gz.

The following dataset was generated:

| Author(s) | Year | Dataset title | Dataset URL | Database and Identifier |
|---|---|---|---|---|
| Harleen Saini | 2019 | Free circular introns with an unusual branchpoint in neuronal projections | https://www.ncbi.nlm.nih.gov/geo/query/acc.cgi?acc=GSE129924 | NCBI Gene Expression Omnibus, GSE129924 |

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
