## [Decision Letter]

Thank you for submitting your article "Free circular introns with an unusual branchpoint in neuronal projections" for consideration by *eLife*. Your article has been reviewed by three peer reviewers, one of whom is a member of our Board of Reviewing Editors, and the evaluation has been overseen by James Manley as the Senior Editor. The following individual involved in review of your submission has agreed to reveal their identity: Peter Sudmant.

The reviewers have discussed the reviews with one another and the Reviewing Editor has drafted this decision to help you prepare a revised submission.

Summary:

Saini and colleagues investigate RNA localization using RNA-Seq analysis of fractionated (cell body and projections) rat hippocampal neurons, with a focus on retained and excised introns. The most interesting finding is a population of stable, tailless lariat-introns in projections that involve non-canonical branch sites. Such RNA species have been reported previously, but not in neuronal projections. Their function, if any, remains unclear. The authors' analyses further indicate that these RNAs lack conservation and ribosome engagement. They propose that these circular introns are splicing by-products and that their accumulation in projections may be a consequence of inefficient debranching and subsequent diffusion. They also observe, as have others, that ribosomal protein mRNAs localize to projections, and propose that this phenomenon, like that of stable intronic circles, may be due to lack of ribosome engagement. While this study does not provide biological insight into stable intronic circles, it suggests how lack of engagement can passively lead to localization in neuronal projections. The study further provides useful RNA-Seq datasets and a resource of differentially localized transcripts that will facilitate future research efforts. As such, it is potentially suitable for publication in the Tools and Resources section of *eLife*. It is requested that the authors address the following comments in a revised manuscript:

Essential revisions:

1) This study is quite descriptive in nature and obviously would be considerably strengthened if the authors could provide functional insight into the stable circular RNAs they detect. While the reviewers appreciate the challenges associated with addressing function – also considering lack of apparent conservation of the lariat circles – it is requested that the authors minimally validate some of their novel findings in Figures 3-6, in particular the localization of intronic circles using smFISH. Moreover, lariat PCR should be performed to confirm the branch points of these RNAs.

2) The reviewers felt that the authors could do much more with their data. For example it is recommended that they correlate differential splicing of exons with differences in RNA localization. Related to this, are the ribosomal protein mRNAs localized to projections representative of distinct splice variants of these transcripts? Previous work has revealed that differential splicing in 5´UTRs can affect ribosome engagement (e.g. PMID 27820807) and may be relevant in this context.

---

## [Author Response]

Essential revisions:1) This study is quite descriptive in nature and obviously would be considerably strengthened if the authors could provide functional insight into the stable circular RNAs they detect. While the reviewers appreciate the challenges associated with addressing function – also considering lack of apparent conservation of the lariat circles – it is requested that the authors minimally validate some of their novel findings in Figures 3-6, in particular the localization of intronic circles using smFISH. Moreover, lariat PCR should be performed to confirm the branch points of these RNAs.

a) We validated projection-enriched (retained and free) and projection-depleted (detained) introns by smFISH. These data are in Figure 4—figure supplement 4. Because we failed to refer to this figure in the main text, however, the reviewers likely were unaware of these data. In the revision, we include a paragraph describing our smFISH observations (subsection “Circular introns with noncanonical branchpoints in projections”, first paragraph).

b) Lariat PCR: Considering that our RNAseq library preparation method uses the same reverse transcription and amplification steps that would be used for lariat PCR, we believe that it would be a redundant control. Instead, in the revision we have added Figure 6—source data 1 that provides details about the chromosomal coordinates of the circular species and their sequence, the location of the circular junction relative to the 3′ splice site, and the number of circularly permuted reads aligning to all free introns labeled in red in Figure 5.

2) The reviewers felt that the authors could do much more with their data. For example it is recommended that they correlate differential splicing of exons with differences in RNA localization. Related to this, are the ribosomal protein mRNAs localized to projections representative of distinct splice variants of these transcripts? Previous work has revealed that differential splicing in 5´UTRs can affect ribosome engagement (e.g. PMID 27820807) and may be relevant in this context.

We agree that these data are a rich resource to address a multitude of questions. For this report, however, we deliberately chose to write a more focused paper, with our focus on analysis of intronic sequences in projections, as this has been a topic of debate in the field for many years. By thoroughly validating these libraries and submitting them to the public repository at NCBI GEO (GSE129924), we hope they will serve as an important resource for the broader community. We also have plans for future analyses ourselves.

Regarding the question about RP mRNA splice variants, we do not observe any differentially expressed alternative isoforms in projections. The differential expression analysis in Figure 2 included all annotated isoforms of RP mRNAs (black dots) and all isoforms expressed by these cells were significantly enriched in projections.